# Hydrogen-Assisted Crack Growth in the Heat-Affected Zone of X80 Steels during in Situ Hydrogen Charging

**DOI:** 10.3390/ma12162575

**Published:** 2019-08-12

**Authors:** Jinglong Qu, Min Feng, Teng An, Zhongnan Bi, Jinhui Du, Feng Yang, Shuqi Zheng

**Affiliations:** 1High Temperature Materials Research Institute, Central Iron & Steel Research Institute, Beijing 100081, China; 2Beijing GAONA Materials & Technology Co., LTD, Beijing 100081, China; 3State Key Laboratory of Heavy Oil Processing and Department of Materials Science and Engineering, China University of Petroleum, Beijing 102249, China

**Keywords:** heat-affected zone, hydrogen embrittlement, electron backscatter diffraction, crack propagation path

## Abstract

Herein, the hydrogen embrittlement of a heat-affected zone (HAZ) was examined using slow strain rate tension in situ hydrogen charging. The influence of hydrogen on the crack path of the HAZ sample surfaces was determined using electron back scatter diffraction analysis. The hydrogen embrittlement susceptibility of the base metal and the HAZ samples increased with increasing current density. The HAZ samples have lower resistance to hydrogen embrittlement than the base metal samples in the same current density. Brittle circumferential cracks located at the HAZ sample surfaces were perpendicular to the loading direction, and the crack propagation path indicated that five or more cracks may join together to form a longer crack. The fracture morphologies were found to be a mixture of intergranular and transgranular fractures. Hydrogen blisters were observed on the HAZ sample surfaces after conducting tensile tests at a current density of 40 mA/cm^2^, leading to a fracture in the elastic deformation stage.

## 1. Introduction

From the viewpoint of having low oil and gas resources and rich coal resources in China, the Chinese government has chosen to develop the coal gas industry. To increase the transport efficiency and reduce costs of natural coal gas, X80 steels with high strength and increased toughness have been widely used for the transport of natural coal gas and cathodic is used to prevent the steel from corrosion [1]. Hydrogen atoms enter the steels in the cathodic-protected process or from coal gas [2,3,4,5,6], leading to a degradation in the mechanical properties of steel such as tensile strength, fracture toughness, and acceleration of the fatigue crack-growth rates [5,7,8,9]. This process is known as hydrogen embrittlement (HE), a topic that requires more attention. 

Several studies have investigated HE and some mechanisms, such as hydrogen pressure mechanism [10], hydrogen-induced localized plasticity (HEIP) [11,12,13], hydrogen-enhanced and strain-induced vacancies [14], hydrogen-enhanced decohesion (HEDE) [15,16], and adsorption-induced dislocation-emission [17,18], have been proposed. The HE failure process generally involves physical adsorption, molecular dissociation, chemical adsorption, and hydrogen diffusion processes. Hydrogen accumulates at the lattice defects, leading to the crack initiation with the combined action of stress and hydrogen. Then, hydrogen diffuses to the tip of the cracks, leading to the acceleration of crack growth, resulting in the mechanical damage of the material. Our previous studies also suggested that an increase in the HE susceptibility is always related to crack initiation and high crack-growth rates [19,20]. To explain the effects of microstructure on the crack growth, the microstructure in the vicinity of the cracks caused by hydrogen needs to be observed using electron backscatter diffraction (EBSD) [21,22,23,24]. Laureys et al. found that hydrogen-assisted martensite decohesion led to the initiation of cracks [25]. Mohtadi-Bonab et al. found crack nucleation and propagation sites caused by hydrogen in X70 steel [26]. After crack initiation, the path of the crack propagation was found to be affected by the grain size, grain boundaries, Taylor factor, and grain orientations. Masoumi et al. revealed that [011] grains increased the hydrogen-induced cracking (HIC) resistance and [001] grains were highly susceptible to HE [22]. Merson et al. determined the path of cleavage cracks by following the [100] planes of grains in low-carbon steel after hydrogen charging [27]. Therefore, the relationship between the microstructure and hydrogen-assisted hydrogen propagation was built in this study by conducting EBSD measurements.

The effect of welding thermal cycles on the microstructure has attracted more attention, since welding is widely used to construct long-distance pipeline steel [28,29]. Microstructure heterogeneity are affected by welding thermal cycle processes, which change the hydrogen diffusion and accumulation [30,31]. Thus, there is a great difference between the base metal and heat-affected zone (HAZ) tested in the hydrogen environment. Zhang et al. analyzed the effects of the HAZ microstructure on the hydrogen permeation of X80 steels, and the results showed that the coarse-grained granular bainite and bainite ferrite in the HAZ increased the hydrogen diffusivity [32]. HAZ showed a higher effective hydrogen diffusivity than the other regions which affected the HE susceptibility [33]. The reduction in the mechanical properties is highly dependent on the synergistic action of microstructure heterogeneity and local hydrogen enrichment. Grain coarsening plays an important role in HE susceptibility [34], with a coarse-grained HAZ having the lowest resistance to HE due to coarsened M/A constituents [28]. Alvaro et al. confirmed that the fracture toughness of the X70 HAZ decreased by 60% in 0.6 MPa hydrogen in comparison with that in air [35]. Our previous study confirmed that the tensile strength and elongation of HAZ samples decreased significantly during in situ hydrogen charging [36]. Some attention has been given to the relation between microstructure evolution and fracturing in the HAZ due to the presence of hydrogen. Nonetheless, the failure mechanism of the HAZ samples during in situ hydrogen charging requires further investigation [28,37].

Herein, the HE susceptibility of HAZ samples prepared using a Gleeble 3500 to heat treat the X80 steel was investigated. Owing to the high diffusivity and low concentration of hydrogen in ferritic steel, the tensile tests were conducted during continuous electrochemical hydrogen charging over a range of current densities. Scanning electron microscopy (SEM) and EBSD measurements were conducted on the fractured surfaces to determine the correlation between the microstructure in the vicinity of the cracks in the HAZ samples’ surface and HE behavior.

## 2. Materials and Methods

### 2.1. Materials and Samples

X80 steel was used in this research with the chemical composition (wt.%) C 0.05, Si 0.22, Mn 1.65, Cr 0.24, Al 0.03, Nb 0.05, Cu 0.12, and Fe balanced. High strength of the X80 steel was attributed to thermomechanical control processing, whereas the better toughness was attributed to the low concentration of carbon and high concentration of microalloying elements in the steel. The samples were put on the X80 steel in transverse orientation with a diameter and thickness of 1219 mm and 22 mm, respectively. The tensile and yield strength of the as-received X80 sample was 655 MPa and 552 MPa, respectively, and the tensile elongation was 25.4%. The surface of the steel in the rolling-transverse direction was observed by optical microscope (OM, Leica, Wetzlar, Germany) after being etched with 4 vol.% nital.

Owing to its high HE susceptibility, coarse-grained HAZ was used as a representative area for the HAZ [37]. The simulated samples were heated up to 1350 ℃, held at that temperature for 5 s, and then cooled to room temperature using a Gleeble 3500 thermomechanical simulator (Dynamic Systems, New York, NY, USA), as shown in Figure 1a. The transformation point in the welding thermal cycles was also measured using Gleeble 3500 apparatus (Dynamic Systems, New York, NY, USA), and the results are shown in Figure 1b. The A_c3_ was approximately 860 °C, which resulted in phase transformation.

### 2.2. Electrochemical Hydrogen Charging and Tensile Testing

The samples used for slow strain-rate tensile testing were machined to have a width of 4 mm (rolling direction), length of 10 mm (transverse direction), and thickness of 1 mm (normal direction). The entering of hydrogen atoms into the steel was induced by an electrochemical hydrogen charging method, which caused severe damage to the steel [36]. The tensile tests were conducted in an aqueous solution of 0.2 mol/L H_2_SO_4_ containing 3 g/L NH_4_SCN at various constant current density of 0, 5, 10, 20, and 40 mA/cm^2^, respectively, with a strain rate of 10^−6^ s^−1^. The hydrogen-charging equipment consists of a VersaSTAT 3F Potentiostat and a three-electrode cell. The sample served as the working electrodes, a saturated calomel electrode was used as the reference electrode, and a platinum wire was the counter electrode. The samples were cathode by setting the current direction. The design of the setup used for in situ tensile under hydrogen-charging was based on the reference [38]. The tensile tests and hydrogen charging were performed simultaneously, since the hydrogen charging time was the same as the tensile testing time. For comparison, the base metal samples were also loaded under the same condition to the identical strain at the current density of 0, 20 and 40 mA/cm^2^.

### 2.3. Microstructural Analysis

The microstructures of the base metal and HAZ were characterized using EBSD (Oxford Instruments, Oxford, UK). The samples for the EBSD observations were mechanically ground using 1200 grit SiC emery paper (Changzhou KingCattle Abrasives Co., ltd., Changhou, China) and then polished with 3 and 1 μm diamond pastes. Finally, the surfaces of the samples were electrochemically polished to remove any residual deformations [39]. The EBSD measurements were conducted with a step size of 0.3 μm. The nanomechanical properties of the samples were characterized using an Agilent G200 nanoindentation system (Agilent Technologies, California, CA, USA), installed with a Berkovich indenter tip, at room temperature. The continuous stiffness mode was used at a constant nominal strain rate of 0.1 s^−1^. The samples were cut at the half of the thickness in the rolling-transverse direction and were electrochemically polished before the indentation measurements. To measure the hardness of a single grain, the maximum load used was 5 mN. Tests were performed 20 times to obtain an average hardness value.

The main objective was to find a correlation between the crystallographic texture and cracking path; therefore, the fractured surfaces were observed by SEM (FEI Quanta 200F) and EBSD. The samples used for the tensile tests were polished using 1 and 0.04 μm diamond paste and colloidal silica suspension, respectively, for 4 h using a VibroMet 2 device. Then, before the EBSD observations were conducted, the samples were prepared by ion-milling, using the Leica EM RES102 apparatus (Leica, Wetzlar, Germany) operated at 3 kV for 10 min after conducting the tensile tests. The recording of high quality EBSD maps requires samples that are flat with a low number of residual deformations. However, X80 steel is a material that has excellent toughness, and mess of strains forms in the material during plastic deformation. Observing the microstructure around the cracks is difficult, due to the formation of necking. Therefore, the EBSD observations was performed on a hydrogen-charged sample at constant current densities of 20 and 40 mA/cm^2^, and the abundance of hydrogen atoms decreased the plastic deformation. 

## 3. Results

### 3.1. Microstructure Evolution

The microstructures of the base metal and HAZ are shown in Figure 2. The X80 steel used herein is a dual-phase steel, wherein the base metal is comprised of polygonal ferrite (PF) with fine grains and granular bainite (GB), which is comprised of fine lath ferrite and some island constituents [40] (Figure 2a). The fraction of PF microstructure in the base metal was about 60 vol.% and the average grain size was estimated to be 8 μm. The GB and PF microstructures were alternated in the steel, which is caused by steel rolling process. The microstructure of the HAZ comprised bainite ferrite (BF), GB, and PF (Figure 2b). Moreover, decreased and increased GB and BF contents, respectively, were observed in the HAZ, and GB was randomly distributed in comparison with the base metal. The fraction of PF microstructure in HAZ was about 80 vol.% and the average grain size was estimated to be 11 μm.

The inverse pole figure (IPF) maps, corresponding image quality (IQ) maps with kernel average micro-orientation (KAM) maps, and grain boundary misorientation distribution of the HAZ microstructure are shown in Figure 3. The analysis results obtained from EBSD are consistent with OM observations, and more crystallographic information was determined through EBSD measurements. The degree of local strain in the GB zone is greater than PF zone owing to the high grain boundary density (Figure 3c,d). A PF grain with a grain size of approximately 10 μm is present as a soft phase due to the low local strain. The GB character has a significant effect on the cracking paths [40] and hydrogen diffusion [41]. GBs are the preferential sites, where hydrogen atoms are trapped, due to the high density of dislocations. Therefore, the base metal shows a lower HE susceptibility than the HAZ [32]. Typical load-displacement curves of the base metal and HAZ are shown in Figure 4. The nanohardness of the HAZ is 2.71 GPa, which is 32% lower than that of the base metal. Increased grain size and local strain, as well as the decreased GB content [42] lead to a decreased hardness of the HAZ samples.

### 3.2. Tensile Properties

Figure 5 shows the stress-strain curves for the HAZ samples tested at a strain rate of 10^−6^ s^−1^ over a series of current densities, which are 0, 5, 10, 20, and 40 mA/cm^2^. The tensile testing and hydrogen charging process were performed simultaneously. For comparison, base metal samples were investigated at the current densities of 0, 20 and 40 mA/cm^2^, as shown in Figure 5a. Results show that the HAZ samples have lower strength and are more ductile than the base metal. Hydrogen decreased the tensile strength and elongation of the base metal and HAZ sample. The HAZ samples have the lower resistance to HE than the base metal samples in the same current density. It was noted that breakage of the HAZ samples occurred before the stress-strain curve reached the yield strength, which was tested at a current density of 40 mA/cm^2^. The decreasing trend of the tensile strength and ductility of the base metal and the HAZ samples with an increase in the current density is shown in Figure 6. Hydrogen-induced degradation of steel is affected by materials, sample geometry, and hydrogen concentration involved. Several studies have reported that the pipeline steels were susceptible to hydrogen embrittlement [43,44]. Herein, the tensile sample used was a sheet with a thickness of 1 mm and hydrogen atoms could easily enter the materials, due to the presence of a hydrogen recombination poisoning agent (NH_4_SCN). Nanninga et al. showed that tensile ductility decreases with an increase in the hydrogen gas pressure, because a high hydrogen gas pressure results in an increased number of hydrogen atoms that can form the surface cracking [45]. Surface cracking due to hydrogen is the main factor behind the reduction in the ductility of the steel. An increase in the current density increases the number of hydrogen atoms that enter into the base metal and the HAZ samples. After the first crack forms in the surface, the crack tip is surrounded by a greater hydrogen concentration due to a high-strain field. The tensile strength of the materials decreased when the hydrogen concentration reached the critical hydrogen concentration [45,46]. A similar trend for the current density and strain rate has been observed for steel [26,47]. Therefore, the tensile strength and ductility of the base metal and the HAZ samples decreased with an increase density.

Fractographic details of the tensile fracture surfaces of the hydrogen-charged HAZ samples at different current densities can be seen in Figure 7. Typical necking was observed in the macroscopic fracture morphologies of the samples tested in air (Figure 7a), with the dominant feature being the ductile dimples. The observed fracture surfaces of the hydrogen-charged HAZ samples at different charging current densities were notably different (Figure 7b–d). Hydrogen significantly reduced the necking in the samples, and fracturing around the root of the surface comprised secondary cracks and flat facets, which are typical HE features. The length and size of the secondary cracks increased with an increase in the current density, which suggests a higher HE susceptibility. Ductile dimples were the dominant feature in the center of the fracture surface.

Figure 8 shows the fracture lateral morphologies of the HAZ samples in the absence and presence of hydrogen. Necking can be observed in the macroscopic fracture surface of the HAZ sample tested in air, as shown in Figure 8a,b. Figure 8c,d show that the surface consists of some cracks that are rather homogeneously distributed. The important issue is the crack propagation path. Figure 8d shows that five or more cracks may join together to yield a longer crack. The fracture features in the presence of hydrogen prove that it induces two main damage modes. Cracks located in the bulk of the samples were visible on the fracture surface of the samples at a constant current of 40 mA/cm^2^ (Figure 7c,d). Brittle circumferential cracks located at the surface of the sample are perpendicular to the loading direction (Figure 8d). The surface cracks in or near the surfaces are attributed to the hydrogen-induced degradation of the steel. Therefore, the failure process of steels can be deduced by studying the paths of the cracks in the fracture surface.

Hydrogen induced cracking (HIC) always initiates at the surface in this steel. Figure 9 shows the typical crack propagation in the surface of the HAZ after tensile tests at a current density of 20 mA/cm^2^. The two crack paths are not linear along a direction. A small crack forms between the two cracks where a high-strain field and two cracks are about to be connected. Figure 9b shows the IPF maps of two cracks that may join together. The same color in IPF maps indicate the same grain orientation. The upper crack propagates through the coarse grain and stops in this grain. The coarse grain, which may be the ferrite grain, reduces both the hydrostatic stress and the effective stress. This increases the tearing resistance and the crack propagation stops. The other crack is due to intergranular cracking, the propagation path of which runs along the grain boundaries. Hydrogen-assisted crack propagation along the special grain boundaries has been reported in many studies [26,27]. Mohtadi-Bonab et al. investigated the crack nucleation and propagation sites of X70 due to hydrogen and showed that that the grains that were oriented for relatively easy slip were subject to intergranular cracking, and those grains tended to be resistant to yielding [26]. The grains that did not yield easily were more prone to transgranular cracking. Dadfarnia et al. found that cracks without hydrogen grew by successive growth and linked the void closest to the crack tip with the crack tip. However, several voids grew and coalesced simultaneously in the presence of hydrogen [48]. Those cracks grew in a stepwise manner [40].

Some blisters were present in the lateral surface of the HAZ sample after tensile tests were conducted at a current density of 40 mA/cm^2^ with sizes of over 0.5 mm in length, as shown in Figure 10. Some cracks were formed at the edge of the blisters, and the corresponding IPF and IQ maps with grain boundary misorientation distributions can be observed in Figure 10c,d, respectively. Blisters may form in the surface after atomic hydrogen diffuses into the steel. Blisters only appear after long hydrogen charging times or at a higher current density. Blisters appeared after the end of the tensile tests and tensile time was approximately 5 h. Tiegel et al. evaluated the formation of cracks and blisters in purified iron with the results showing that the hydrogen partial pressure that resulted in blister formation was approximately 1800 MPa, with the pressure in cracks measured to be 50 MPa [10]. Therefore, the hydrogen partial pressure in the blisters generates high localized stress (Figure 10a,b), which initiates cracking along the lines of weakness in the steel. Elboujdaini et al. showed that blisters often occur when the crack is unable to further propagate inside the steel [49]. Figure 10c also shows a mixture of intergranular and transgranular fractures and larger cracks formed by the coalescing of small cracks. Therefore, the formation of hydrogen blisters leads to earlier failure prior to the plastic yield deformation of the sample.

## 4. Discussion

The degree of HE susceptibility of materials can be quantified by measuring different parameters, such as reduction in area, fatigue crack-growth rate, fatigue life, fracture toughness, and time to failure [31,50]. Reduction in the tensile strength and elongation suggests that the base metal samples and the HAZ samples of X80 steel were strongly susceptible to HE [3,51], as shown in Figure 5, with a general trend that the HE susceptibility increases with an increase in the hydrogen concentration. Figure 6 shows that the hydrogen concentration increases because of increasing hydrogen pressure or current density and reaction time (fatigue frequency or strain rate) [52,53,54,55,56].

The HE susceptibility of the materials generally increased with an increase in the steel strength [57]. The HE susceptibility of the HAZ samples was higher than that of the base metal samples. The difference in microstructure seemed to be the main factor that affected hydrogen trapping and diffusion [58]. The electrochemical permeation technique was the available method to study the relationship between hydrogen diffusion and trapping behavior and bainite microstructure. Park et al. evaluated hydrogen trapping efficiency in different microstructures of pipeline steel and found that acicular ferrite and bainite can act as hydrogen traps, with the trapping efficiency of acicular ferrite being higher than that of other microstructures [59]. Arafin et al. confirmed that the bainite lath type microstructure played a key role in causing the embrittlement compared with the GB microstructure [60]. Furthermore, GB, which consists of fine lath ferrite and some island M/A constituents, exhibits a high trapping efficiency. The GB microstructure has a high-angle grain boundary density and grain boundary mismatch can also influence the hydrogen cracking behavior. Therefore, a higher fraction of granular bainite in X80 steels increased sub-surface hydrogen concentration and exhibited the lower apparent hydrogen diffusivity [61]. Compared with the base metal, the HAZ containing lower fraction of granular bainite had lower concentration of reversible traps and consequent lower diffusible hydrogen, resulting in higher diffusion kinetics of hydrogen atoms. Meanwhile, the content of the GB microstructure is generally associated with the strength and hardness of the X80 steel, as shown in Figure 4 and Figure 5. Zhao et al. showed that the hydrogen diffusivity of a simulated HAZ was higher than that of a base metal, which indicated a higher HE susceptibility [37]. Alvaro et al. investigated the HE susceptibility of the HAZ of X70 steel with the results confirming that the critical hydrogen pressure for the onset of HE was between 0.1 and 0.6 MPa, which was lower than that of the base metal [35]. The above discussion confirmed that HAZ has a low GB ratio, which increases the hydrogen diffusivity rate in comparison with that of the base metal. The HE susceptibility of the HAZ is higher than that of the base metal, as shown in Figure 5. Koyama et al. [62] reviewed the recent progress in the microstructure-specific hydrogen mapping techniques and many important results were obtained. The relation between the microstructure of the HAZ and HE susceptibility requires further study by using those advanced techniques.

Recent studies have focused on establishing how HE mechanisms trigger a fracture, resulting in material failure [48]. The mechanisms of HIC can be confirmed by fractographic observations; however, each of the cracking mechanisms that have been proposed is controversial. The mechanism is different for every individual case because of the different material, environment, and testing conditions used. For brittle fractures, the HEDE theory governs the failure process and for ductile fractures the hydrogen-induced localized plasticity (HELP) theory promotes the strain localization that leads to fracturing. Neeraj et al. proposed the plasticity-generated, hydrogen-stabilized vacancy damage accumulation and nanovoid coalescence mechanism for steels by analyzing the microstructure of the fracture surface [63]. Combining HELP with the HEDE and hydrogen pressure theories may lead to a better explanation of the stress-oriented HIC. Based on the HELP mechanism, hydrogen accumulation at key microstructural features such as grain boundaries results in the localization of enhanced dislocation mobility. Crack initiation always occurs in structural defects, but the influence of hydrogen on crack initiation was not studied here. After crack initiation, hydrogen accumulates at the plastic zone of the crack tip, and the crack path is notably different. Cracking can occur in a wide range of grain orientations with detailed EBSD analysis showing that some HIC was observed, as shown in Figure 9 and Figure 10. The fracture mode was a mixture of intergranular and transgranular fractures. Therefore, cracks can be bridge-linking microscopic and macroscopic failures in steel. The mechanisms of the hydrogen-induced degradation of steel can be researched by studying the cracking path.

An increase in the HE susceptibility of HAZ can be attributed to high crack-growth rates during tensile testing. The cracking process generally comprises three stages: (a) hydrogen causes cracks to initiate, (b) crack initiates, and (c) crack propagates until failure occurs. Crack initiation due to hydrogen always occurs in structural defects and nonmetallic inclusions [64]. HIC behavior was unaffected by the applied tensile or residual stress; however, hydrogen accumulates under a high stress field. HIC may occur in the presence of tensile stress, which is called stress-oriented HIC. The cracks in stress-oriented HIC or HIC were notably different in terms of type and propagation path. The crack path herein was oriented perpendicular to the principal applied stress, as shown in Figure 8d. The fracture mode was a mixture of intergranular and transgranular fractures. Fracturing may occur around the grain boundaries (intergranular fracture) because the grain boundaries can act as trapping sites due to the existence of a density of dislocations [65]. The HELP mechanism for embrittlement seems to act when the crack also propagates in the ferrite phase (Figure 9b). The void growth and linkage cause the crack tips of developing cracks to connect. When hydrogen is present, several voids grow and coalesce [48]. Therefore, the crack path with respect to the microstructure can be observed in Figure 9b and Figure 10c. In terms of intergranular cracking, large grain boundary misorientations may be the preferred sites, which are the grains of the GB microstructure. Cracks that propagated in the ferrite phase were transgranular cracks and stopped in the vicinity of grains, as shown in Figure 10c. Zhao et al. also proved that ferrite-ferrite and ferrite-bainite boundaries in X80 steels can efficiently slow crack propagation, showing that cracks grew in a stepwise manner [40]. Therefore, the stepwise cracks were found to be the result of the comprehensive effect of hydrogen, stress, and microstructure.

It was noted that hydrogen blisters formed on the surface and the cracks were located in the interior of the material [10]; therefore, hydrogen blisters and hydrogen-induced cracks have always been discussed separately. This study shows that the blisters and cracks appear concurrently in the fracture surfaces of the HAZ samples, and that tensile stress promoted the formation of blisters and cracks were located in the interfaces between the blister and matrix metal. The formation of blisters was dependent on the time and current density of the hydrogen charging. Figure 10 shows that longer time periods (over 5 h) and higher current density (40 mA/cm^2^) of hydrogen charging led to the formation of blisters in the surface of the HAZ samples before the tensile curve reached the yield stage. The interaction of the localized plastic deformation and the accumulation of hydrogen atoms at the interfaces enhanced the deformation of cracking, which is in agreement with the HELP mechanism. Therefore, the damage due to blisters and hydrogen-induced cracks was discussed herein and the failure mechanism was found to be consistent for the HAZ samples.

## 5. Conclusions

The main objectives of this study were to establish the link between the crystallographic texture and crack path of the HAZ samples. Tensile tests were performed on the HAZ samples during in situ hydrogen charging over a series of current densities. The crack paths in the fracture surfaces of the HAZ samples were investigated using EBSD. The following conclusions were drawn:The HE susceptibility of the base metal samples and the HAZ samples increased with an increase in the current density. The fracture surfaces showed that hydrogen significantly reduced necking and the surface of the HAZ was comprised of secondary cracks and flat facets.The HE susceptibility of the HAZ sample was higher than that of the base metal samples in the same current density because of the lower ratio of GB microstructure in the HAZ. GBs are the preferential trapping sites due to the high density of dislocation which affect the hardness and the HE susceptibility.The crack propagation path showed that five or more cracks may join together to form long cracks. The fracture morphologies were found to be a mixture of intergranular and transgranular fractures.The formation of hydrogen blisters in the HAZ tensile samples at a hydrogen charging current density of 40 mA/cm^2^ led to early failure of the material prior to the plastic yield deformation of the sample.

## Figures and Tables

**Figure 1 materials-12-02575-f001:**
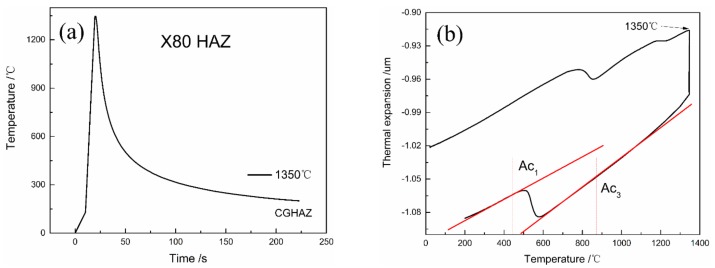
Schematic of the (**a**) welding thermal cycles and (**b**) transformation point.

**Figure 2 materials-12-02575-f002:**
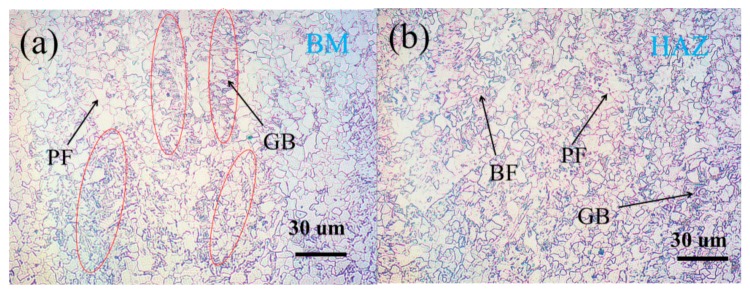
Microstructure of the X80 steel observed by optical microscope (OM) in the rolling-transverse direction: (**a**) the base metal (BM)and (**b**) heat-affected zone (HAZ).

**Figure 3 materials-12-02575-f003:**
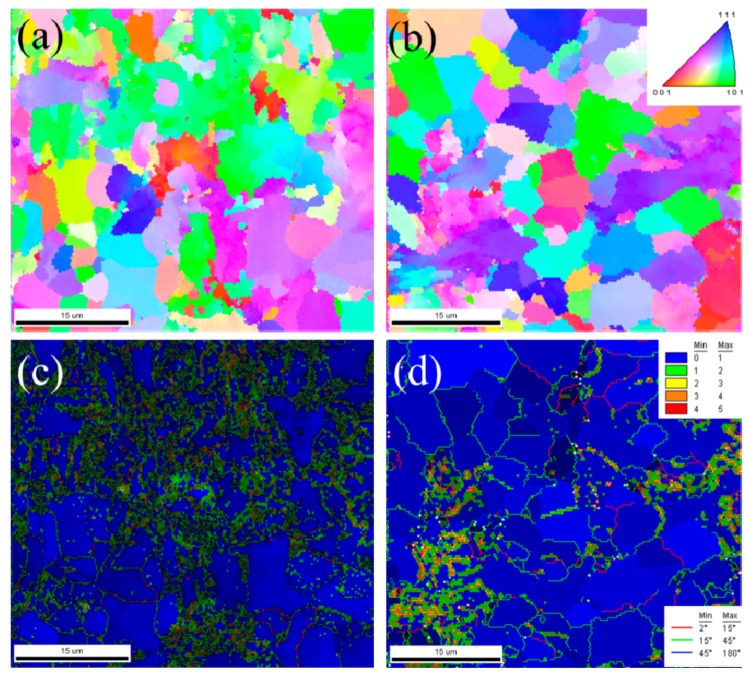
Inverse pole figure (IPF) maps of the (**a**) BM and (**b**) HAZ. image quality (IQ) maps with kernel average micro-orientation (KAM) maps and grain boundary misorientation distribution of the (**c**) BM and (**d**) HAZ.

**Figure 4 materials-12-02575-f004:**
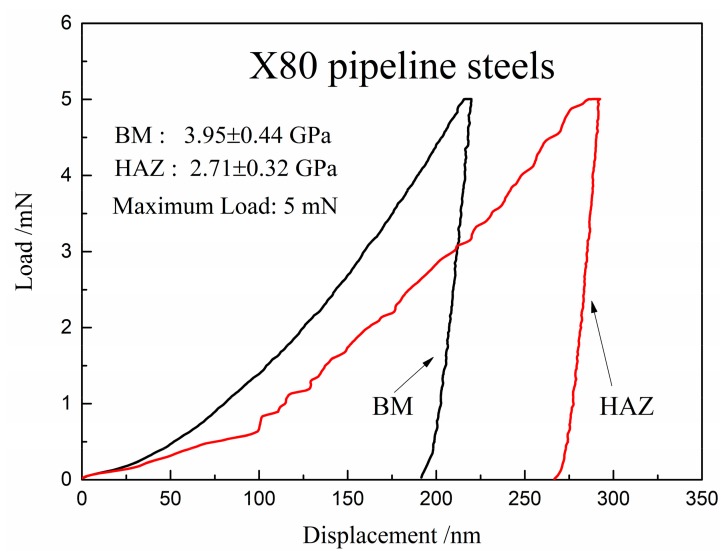
The displacement-load curves of the X80 base metal and HAZ.

**Figure 5 materials-12-02575-f005:**
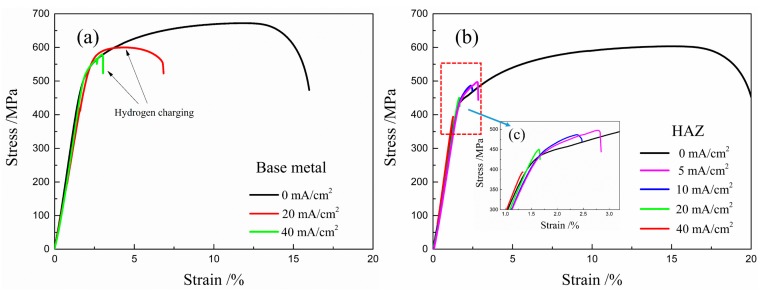
Stress-strain curves for the X80 steel: (**a**) the base metal samples and (**b**) the HAZ samples tested during hydrogen charging at different current densities of 0, 5, 10, 20, and 40 mA/cm^2^.

**Figure 6 materials-12-02575-f006:**
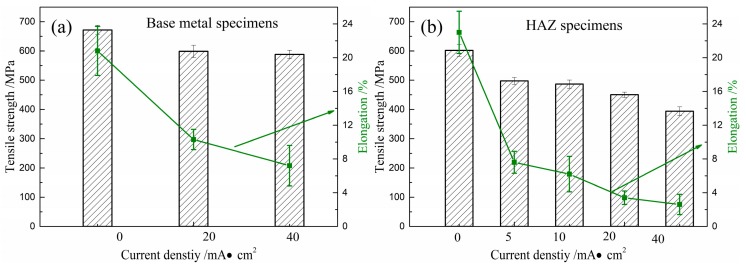
The trend in the tensile strength and elongation of the base metal samples (**a**) and the HAZ samples (**b**) at failure during hydrogen charging at different current densities.

**Figure 7 materials-12-02575-f007:**
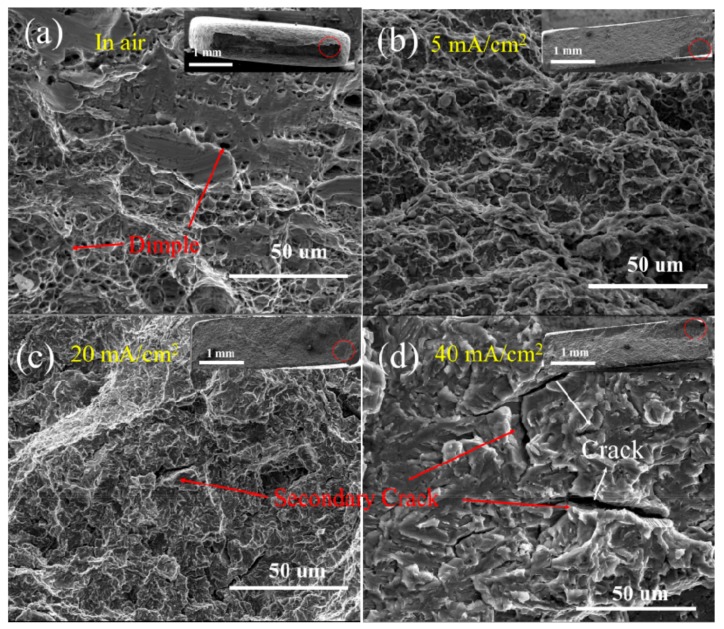
The fracture surface of the HAZ samples tested in (**a**) air and during in situ hydrogen charging at different current densities of (**b**) 5, (**c**) 20, and (**d**) 40 mA/cm^2^.

**Figure 8 materials-12-02575-f008:**
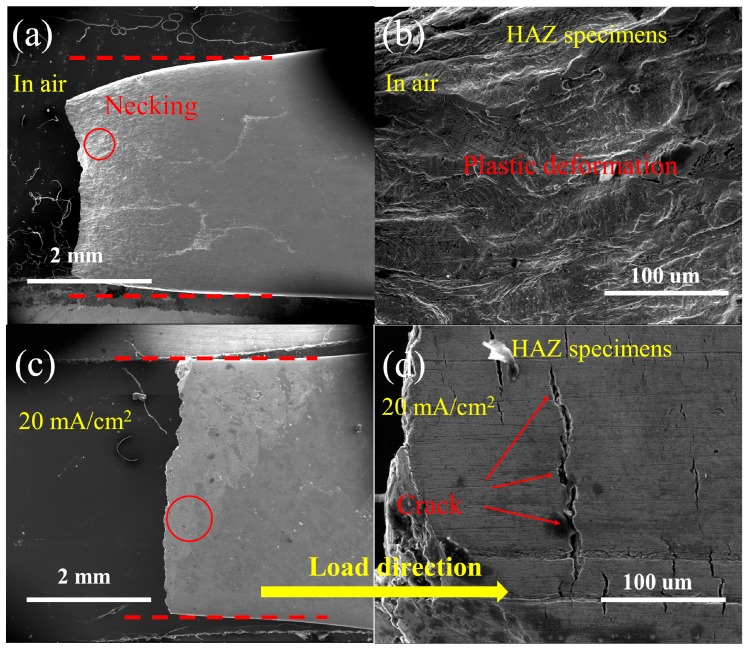
The morphology of the lateral morphologies of the heat-affected zone (HAZ) (**a**,**b**) in air and (**c,d**) during in situ hydrogen charging at a current density of 20 mA/cm^2^.

**Figure 9 materials-12-02575-f009:**
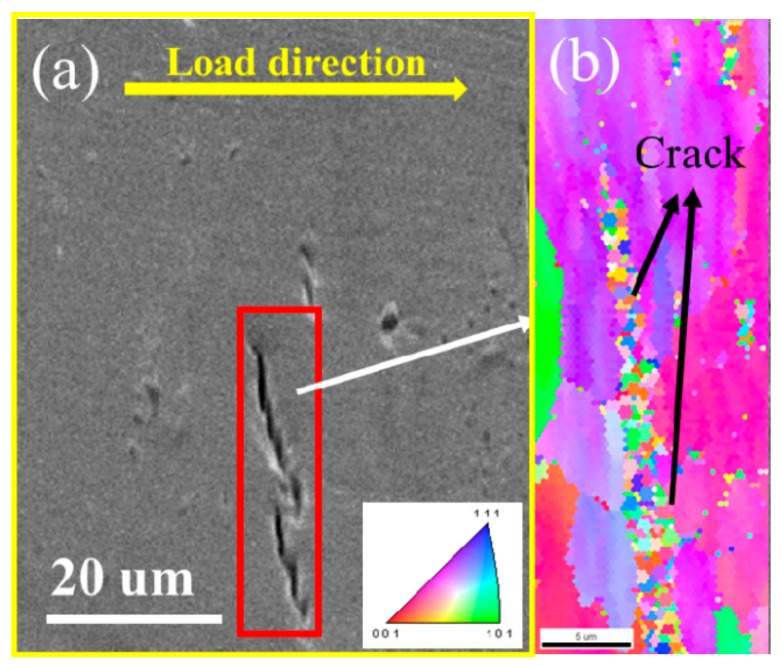
Microstructure of the lateral surface of a HAZ sample in the vicinity of the hydrogen-induced cracking (HIC): (**a**) Scanning electron microscopy (SEM)image and (**b**) The inverse pole figure (IPF)map.

**Figure 10 materials-12-02575-f010:**
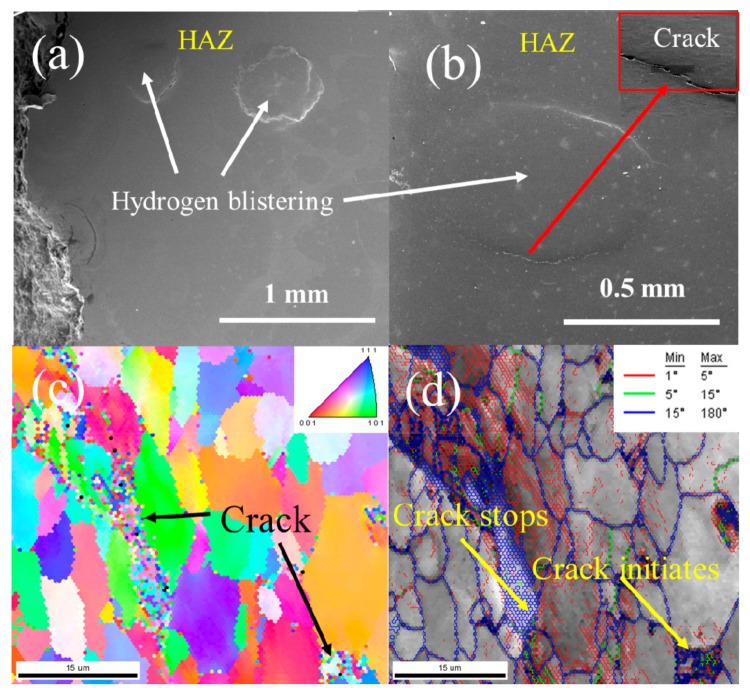
Hydrogen blistering and a crack on the surface of the HAZ samples after tensile tests at a current density of 40 mA/cm^2^: (**a**,**b**) SEM image of the hydrogen blistering; (**c**) IPF map of the crack, and (**d**) IQ maps with the grain boundary misorientation distribution of the cracks.

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
