# Peer review of "Hydrogen-Assisted Crack Growth in the Heat-Affected Zone of X80 Steels during in Situ Hydrogen Charging"

_materials, 2019, doi:10.3390/ma12162575_

Round 1
Reviewer 1 Report
Please see my comments.

Reviewer 2 Report
Paper deals with important subject of hydrogen embrittlement of steel. There are some major issues that need addressing given but not limited to itemised list below:
- Introduction is very poor and needs to be rechecked and corrected. References for AIDE mechanism are missing.
- Experimental methods give only vague description of nanoindentation procedure. Here full description of measurement method is missing and used indentation tip type. Some words are misspelled and units are not written correctly. For example Nital used was 4 vol.% not 4 %. strain rate is given as 10^{-6} /s instead 10^{-6} s^{-1}. Observed blisters appeared due to high charging current as correctly later discussed.
- In results nothing is mentioned on Mn segregations expected in this type of steel. Grain size needs to be give also from optical microscopy comparing differences between HAZ and base microstructure. Some abbreviations used were not explained. For example, KAM.
In tensile properties section is a lot of speculation. Also NH4SCN is not hydrogen poison, but is hydrogen recombination poison. Section on HIC and description of Fig. 9 also needs to be rewritten and strengthen with more evidence.
- If you named section 3 as results and discussion I wonder why you have separate section 4 with name discussion.
Author Response
Paper deals with important subject of hydrogen embrittlement of steel. There are some major issues that need addressing given but not limited to itemised list below.
Answers: Thanks for your comments and we have revised our paper carefully. We hope your questions have been clearly explained and our efforts meet with your approval. We will answer you point by point as follows.
Comment 1:
(1) Introduction is very poor and needs to be rechecked and corrected. (2) References for AIDE mechanism are missing.
Answers: (1) Thank you for your suggestion, we have rechecked and corrected the introduction section in the manuscript and revised portions are marked with underlines and red colors.
(2) References [17, 18] for AIDE mechanism have been added in this manuscript.
[17] Lynch S.P. Environmentally assisted cracking: Overview of evidence for an adsorption-induced localised-slip process. Acta Mater. 1988, 36, 2639-2661.
[18] 18. Lynch S.P. Metallographic contributions to understanding mechanisms of environmentally assisted cracking. Mater. Char. 1989, 23, 147-171.
Comment 2:
(1) Experimental methods give only vague description of nanoindentation procedure. Here full description of measurement method is missing and used indentation tip type. (2) Some words are misspelled and units are not written correctly. For example Nital used was 4 vol.% not 4 %. strain rate is given as 10^{-6} /s instead 10^{-6} s^{-1}. (3) Observed blisters appeared due to high charging current as correctly later discussed.
Answers: (1) Thank you for your suggestion, and we are very sorry for our imprecise writing. The indentation details “The nanomechanical properties of the samples were characterized using an Agilent G200 nanoindentation system installed with a Berkovich indenter tip at room temperature. The continuous stiffness mode was used at a constant nominal strain rate of 0.1 s-1. The samples were cut at the half of the thickness in the rolling-transverse direction and were electrochemically polished before the indentation measurements.” have been added in this manuscript.
(2) “4 %” has been reword as “4 vol.%”, “10-6 /s” has been reword as “10-6 s-1”.
(3) The sentence “It was noted that hydrogen blisters appeared on the surface of the samples at a constant current density of 40 mA/cm2, and the cracks around the blisters were also observed.” has been deleted in this manuscript and this sentence was a result not the experiment method.
Comment 3:
(1) In results nothing is mentioned on Mn segregations expected in this type of steel. (2) Grain size needs to be give also from optical microscopy comparing differences between HAZ and base microstructure. (3) Some abbreviations used were not explained. For example, KAM.
Answers: (1) Thank you for your suggestion, Mn segregations appeared in this type of pipeline steel, however, sha et al. [xx] investigated the HIC of X80 steel with reduced Mn content, the results showed that the X80 steel with reduced Mn content decreased the content of MnS which is suitable for sour service application. Meanwhile, the Mn segregations in base metal and HAZ were not different and was not mentioned in this manuscript.
[xx] Sha, Q.; Li, D., Microstructure, mechanical properties and hydrogen induced cracking susceptibility of X80 pipeline steel with reduced Mn content. Materials Science and Engineering: A 2013, 585, 214-221.
(2) The fraction of PF microstructure in base metal was about 60 vol.% and the average grain size was estimated to be 8 mm. Decreased and increased GB and BF contents, respectively, were observed in the HAZ. The fraction of PF microstructure in HAZ was about 80 vol.% and the average grain size was estimated to be 11 mm.
(3) The abbreviation used was explained as “kernel average microrientation (KAM)”.
Comment 4:
(1) In tensile properties section is a lot of speculation. (2) Also NH4SCN is not hydrogen poison, but is hydrogen recombination poison. (3) Section on HIC and description of Fig. 9 also needs to be rewritten and strengthen with more evidence.
Answers: (1) Thank you for your suggestion, the tensile properties section have been rechecked and rewritten some expressions in this manuscript which are marked with underlines and red colors.
(2) The NH4SCN is the hydrogen recombination poison, and we have corrected this inappropriate expression in this manuscript.
(3) Section of HIC and descript of Figure 9 have be rewritten in this manuscript. “A small crack forms between the two cracks where own a high-strain field and two cracks are about to be connected. Figure 9b shows IPF maps of two cracks that may join together. The same color in IPF maps indicate the same grain orientation. The upper crack propagates through the coarse grain and stops in this grain. The coarse grain which may be the ferrite grain reduces both the hydrostatic stress and the effective stress. This increases the tearing resistance and the crack propagation stops. The other crack is due to intergranular cracking, the propagation path of which runs along the grain boundaries. Hydrogen-assisted crack propagation along the special grain boundaries have been report in many research [26, 27]. Mohtadi–Bonab et al. investigated the crack nucleation and propagation sites of X70 due to hydrogen and showed that that the grains that were oriented for relatively easy slip were subject to intergranular cracking, and those grains tended to be resistant to yielding [26]. The grains that did not yield easily were more prone to transgranular cracking. Dadfarnia et al. found that crack without hydrogen grew by successive growth and linkage of the void closest to the crack tip with the crack tip. However, several voids grew and coalesced simultaneously in the presence of hydrogen [48]. Those cracks grew in a stepwise manner [40].”
Comment 5:
If you named section 3 as results and discussion I wonder why you have separate section 4 with name discussion.
Answers: Thank you for your suggestion, the name sections 3 as “results and discussion” has been reworded as “3. Results”
Reviewer 3 Report
The manuscript is written nicely with sufficient scientific proof for the findings and claims. The images, especially the micrographs can be improved for the visibility of the text in there. It is very hard to read the text in the current form.
Author Response
The manuscript is written nicely with sufficient scientific proof for the findings and claims. The images, especially the micrographs can be improved for the visibility of the text in there. It is very hard to read the text in the current form.
Answers: Thank you for your comments and we are very glad that our efforts have won your approval. The size of images had adjusted for your review in the manuscript.
Round 2
Reviewer 1 Report
Please see my commentts.

Reviewer 2 Report
Authors followed all suggestions.
Author Response
Comment: Authors followed all suggestions.
Answers: We are very glad that our efforts meet your approval. I would like to thank you again for your careful and constructive reviews.